

# High efficiency of livestock ammonia emission controls on alleviating particulate nitrate during a severe winter haze episode in northern China

**Zhenying Xu[1], Mingxu Liu[1], Yu Song[1*], Shuxiao Wang[2*], Lin Zhang[3], Tingting Xu[1], Tiantian Wang[1], Caiqing Yan[1], Tian Zhou[1], Yele Sun[4], Yuepeng Pan[4], Min Hu[1], Mei Zheng[1*] and Tong Zhu[1]**

[1]State Key Joint Laboratory of Environmental Simulation and Pollution Control, Department of Environmental Science, Peking University, Beijing, 100871, China

[2]State Key Joint Laboratory of Environment Simulation and Pollution Control, School of Environment, Tsinghua University, Beijing 100084, China

[3]Laboratory for Climate and Ocean-Atmosphere Studies, Department of Atmospheric and Oceanic Sciences, School of Physics, Peking University, Beijing 100871, China

[4]State Key Laboratory of Atmospheric Boundary Layer Physics and Atmospheric Chemistry, Institute of Atmospheric Physics, Chinese Academy of Sciences, Beijing 100029, China

[*]Corresponding author: Yu Song (songyu@pku.edu.cn), Shuxiao Wang (shxwang@tsinghua.edu.cn), Mei Zheng (mzheng@pku.edu.cn).



**Abstract**

Although nitrogen oxide (NOx) emission controls have been implemented for several
years in northern China, recent observations show particulate nitrate ($NO_3^-$) is becoming
increasingly important during haze episodes. In this study, we find that particulate $NO_3^-$
formation would easily become $NH_3$-limited under severe haze conditions, enhancing its
sensitivity to $NH_3$ emission controls. Furthermore, improved manure management of
livestock husbandry could reduce 40% of $NH_3$ emissions (currently 100 kiloton per a
month) in winter of northern China. Under this emission reductions scenario, simulations
from the thermodynamic equilibrium model (ISORROPIA-II) and the Weather Research
and Forecast model coupled chemistry (WRF-Chem) all show that particulate $NO_3^-$ could
be reduced by approximately 40% during a typical severe haze episode (averagely from
40.8 to 25.7 $\mu g/m^3$). Our results indicate that reducing livestock $NH_3$ emissions would be
highly effective to reduce particulate $NO_3^-$ during severe winter haze events.

**1 Introduction**

In northern China (including Beijing, Tianjin, Hebei, Shandong, Shanxi and Henan),
severe haze pollution events occur frequently during wintertime, with the concentration of
$PM_{2.5}$ (particles with an aerodynamic diameter less than 2.5 μm) reaching hundreds of
micrograms per cubic meter (Wang et al., 2015;Zheng et al., 2015;Elser et al., 2016). In
severe haze events, secondary inorganic aerosol (SIA) plays a crucial role in haze formation,
accounting for 30–77% of $PM_{2.5}$ (Huang et al., 2014). In October 2014, four extreme haze
episodes were reported in North China Plain (NCP), with the concentrations of $PM_{2.5}$
exceeding 400 $\mu g/m^3$, and the concentrations of SNA (sulfate, nitrate, and ammonium) and
particulate nitrate ($NO_3^-$) at this time exceeding 190 and 90 μg/m, respectively(Yang et al.,
56 2015).

To mitigate severe fine particle pollution, the Chinese government has been taking
strong measures to control $SO_2$ and $NO_x$ emissions. It has been reported that $SO_2$ emissions
in China have been reduced by 75% since 2007 (Li et al., 2017a). Meanwhile, particulate
sulfate has also been found to decrease since 2005 (Geng et al., 2017). Liu et al. (2017a)
found that NOx emissions in 48 Chinese cities decreased by 21% from 2011 to 2015.
Unfortunately, in recent years, no obvious decreasing trend in the concentration of
particulate $NO_3^-$ had been observed in northern China (Zhang et al., 2015;Li et al., 2017b)
and the ratio of ammonium nitrate ($NH_4NO_3$) to SNA increased continuously during severe
haze events (Li et al., 2017d;Yang et al., 2017).
It was reported in recent studies that the large atmospheric $NH_3$ emissions in northern
China have made particulate $NO_3^-$ become more important, and it will reduce the
effectiveness of existing $PM_{2.5}$ control strategies through $SO_2$ and $NO_X$ emission reductions
(Wang et al., 2013;Fu et al., 2017). However, the effectiveness of controlling $NH_3$
emissions in reducing particulate $NO_3^-$ during severe winter haze events has not been
reported.
In this study, we firstly compile a comprehensive $NH_3$ emission inventory for northern
China in winter of 2015, and estimate the $NH_3$ emission reductions by improving manure
management. Then, the ISORROPIA-II and WRF-Chem models are used to investigate the



effectiveness of $NH_3$ emission reductions on alleviating particulate $NO_3^-$ during a severe
haze episode. Finally, the molar ratio of observational data is used to explore the particulate
$NO_3^-$ reductions efficiency during the wintertime.

## 2 Methods and Materials

### 2.1 Observational data

Hourly time-resolution aerosol and gas measurements were conducted at the Peking
University urban atmosphere environment monitoring station (PKUERS) (39.991N,
116.313E) in Beijing in December 2015 and December 2016. A commercialized semi-
continuous In-situ Gas and Aerosol Composition (IGAC) Monitor was used to measure the
concentrations of water-soluble ions (e.g., $NH_4^+$, $SO_4^{2-}$, $NO_3^-$, $Na^+$, $K^+$, $Ca^{2+}$, $Mg^{2+}$, $Cl^-$) in
$PM_{2.5}$ and inorganic gases (e.g., $NH_3$, $HNO_3$, $HCl$). Relative humidity (RH) and
temperature were observed at 1-min resolution at the same site. The quality assurance and
control for the IGAC was described in Liu et al. (2017b). A typical severe haze episode
occurred during the 6 to 10 in December 2015, with daily average concentrations of $PM_{2.5}$
exceeding 150 μg/m$^3$ for three days ($PM_{2.5}$ data are from China National Environmental
Monitoring Centre). The average RH and temperature in this haze event were 60.9 ± 11.4%
and 276.5 ± 1.4 K. The south wind was dominant with wind speed mostly less than 3 m/s.
The average concentrations of particulate $NO_3^-$, $NH_4^+$ and $SO_4^{2-}$ were 39.8 ± 14.7 μg/m$^3$,
27.7 ± 8.6 μg/m$^3$ and 42.4 ± 16.0 μg/m$^3$, respectively. The ratios of particulate $NO_3^-$
concentrations to SNA were 36.5 ± 4.0%.

### 2.2 $NH_3$ emission inventory

A comprehensive $NH_3$ emission inventory in 2015 at a monthly and 1 km × 1 km
resolution is developed based our previous studies (Huang et al., 2012;Kang et al., 2016).
A diverse range of sources, including both agricultural (livestock manure and chemical
fertilizer) and non-agricultural sectors (e.g., traffic, biomass burning etc.) were fully
considered. Recent studies documented that our results agreed well with the satellite
measurements by Infrared Atmospheric Sounding Interferometer (IASI) (Van Damme et
al., 2014) and Tropospheric Emission Spectrometer (TES), and inverse model results by
using ammonium ($NH_4^+$) wet deposition data (Paulot et al., 2014;Zhang et al., 2018).
According to our inventory, the estimated $NH_3$ emission amount in northern China was 100
kiloton in December 2015. The largest source was livestock waste (57.0% of the total
emissions), following by vehicle (12.2%), chemical industry (8.8%), biomass burning
(5.4%), waste disposal (4.0%), synthetic fertilizer applications (2.4%) and other minor
sources (9.1%). The proportion of chemical fertilizer is very small due to the limited
fertilization activity in winter.

### 2.3 ISORROPIA-II and WRF-Chem models

The thermodynamic equilibrium model, ISORROPIA-II (Fountoukis and Nenes,
2007), being used to determine the phase state and composition of an $NH_4^+$ - $SO_4^{2-}$ - $NO_3^-$ -
$K^+$ - $Ca^{2+}$ - $Mg^{2+}$ - $Na^+$ - $Cl^-$ - $H_2O$ aerosol system with its corresponding gas components
in thermodynamic equilibrium, was used to investigate the response of particulate $NO_3^-$ to
$NH_3$ emission reductions. Using measurements of water-soluble ions, T and RH from
PKUERS as inputs, ISORROPIA-II can avoid the inherent uncertainty in estimates of



emission inventories, pollutant transport, and chemical transformation. In this study,
ISORROPIA-II was run in the "forward mode" and assuming particles are "metastable"
with no solid precipitates, which is due to the relatively high RH range observed during
this haze event (RH = 60.9 ± 11.4%).
We assess the performance of ISORROPIA-II by comparing measured and predicted
particulate $NO_3^-$, $NH_4^+$ and gaseous $HNO_3$, $NH_3$. An error metric, the mean bias (MB), is
used to quantify the bias (the description of MB is shown below Figure S1). The predicted
particulate $NO_3^-$, $NH_4^+$ and $NH_3$ agree well with the measurements and the value of $R^2$ are
0.99, 0.94 and 0.84, respectively (Figure S1). The MB is only 1.0 ± 1.1 μg/m³, 0.3 ± 1.3
μg/m³ and -1.8 ± 1.6 μg/m³, respectively. However, the model performs poorly on $HNO_3$,
with an $R^2$ of only 0.06 and a MB of -1.0 ± 1.1 μg/m³. This is because particulate $NO_3^-$ is
predominantly in the particle phase (the mass ratio of particulate $NO_3^-$ to the total nitric
acid (TN = $NO_3^-$ + $HNO_3$) was 99.2 ± 1.9%), small errors in predicting particulate $NO_3^-$
are amplified in $HNO_3$ predicting. Since the MB of $HNO_3$ is much smaller than the
observed particulate $NO_3^-$ (39.8 ± 14.7 μg/m³) and $NH_4^+$ (27.7 ± 8.6 μg/m³), this bias have
little influence on simulating the efficiency of particulate $NO_3^-$ reductions.
In the real atmosphere, changes in the level of total ammonia (TA = $NH_4^+$ + $NH_3$) can
affect the lifetime of TN (Pandis and Seinfeld, 1990). This is because the gaseous $HNO_3$
has a faster deposition rate in the atmosphere than particulate $NO_3^-$, and reductions in $NH_4^+$
may prompt particulate $NO_3^-$ partitioning into the gas phase. In such a case, the
concentration of TN would not remain constant but decrease. In order to consider these,
we use the Weather Research and Forecast Model coupled Chemistry (WRF-Chem) model
(ver. 3.6.1) to investigate the effect of $NH_3$ emission controls on particulate $NO_3^-$ formation
in the regional scale. The simulations were performed for the severe haze event during 6 to
10 December 2015. The modeling domain covered the whole northern China with
horizontal resolution of 25 km and 24 vertical layers from surface to 50 hPa. The initial
meteorological fields and boundary conditions were taken from the 6 h National Centers
for Environmental Prediction (NCEP) global final analysis with a 1° × 1° spatial resolution.
The inorganic gas-aerosol equilibrium was predicted by Multicomponent Equilibrium
Solver for Aerosols (MESA) in WRF-Chem(Zaveri et al., 2005;Zaveri et al., 2008). The
Carbon-Bond Mechanism version Z (CBMZ) photochemical mechanism and Model for
Simulating Aerosol Interactions and Chemistry (MOSAIC) aerosol model were used in this
study(Fast et al., 2006). Anthropogenic emissions from power plants, industrial sites,
residential locations, and vehicles were taken from the Multi-resolution Emission
Inventory for China (MEIC; available at www.meicmodel.org).The WRF-Chem model
could approximately reproduce the temporal variations of inorganic aerosol components in
this haze event (Figure S2).

**3 Results**
**3.1 High potential reduction of wintertime $NH_3$ emissions in northern China**
Livestock husbandry accounts for the largest proportion of $NH_3$ emissions in winter
of northern China (approximately 60%), which is mainly caused by the poor manure
management. There are three main animal-rearing systems in China: free-range, grazing
and intensive. On the one hand, the proportion of intensive livestock husbandry in China



is only about 40%, far lower than that of developed countries. As a result, the widespread
free-range and grazing animal rearing systems contribute more than half of the total
livestock $NH_3$ emissions due to lacking manure collection and treatment (Kang et al., 2016).
On the other hand, there were no relevant regulations about storage and application of
manure for intensive farms in China in the past few decades. This causes most livestock
farms also lack necessary measures and facilities for manure collection and storage
(Chadwick et al., 2015). Meanwhile, most of the solid fraction of manure is applied to
crops without any treatment and the liquid fraction is often discharged directly (Bai et al.,
2017).
Due to the current poor manure management in China, the improved manure
management may have great potential for $NH_3$ emission reductions from livestock
husbandry (Wang et al., 2017). The improved manure management mainly includes three
phases: in-house handling, storage and land application (Chadwick et al., 2011). According
to previous studies, for in-house handling, regularly washing the floor and using slatted
floor or deep litter to replace solid floor could both reduce $NH_3$ emissions by more than 50%
(Monteny and Erisman, 1998;Hou et al., 2015). For storage, covering slurry and manure
could reduce $NH_3$ emissions by about 50%-70% (Hou et al., 2015;Wang et al., 2017). For
land application, cultivating the soil surface before application or incorporation and
injection could both reduce $NH_3$ emissions by more than 50% (Sommer and Hutchings,
2001;Hou et al., 2015).
Based on the above research results, the livestock $NH_3$ emission reductions strategies
applied in this study include the following steps. Firstly, the proportion of intensive
livestock production was raised from 40% to 80% in our $NH_3$ emission inventory model.
In our model, the animals in free-range and grazing animal rearing systems are assumed to
live outdoors for half a day, and the improved manure management is only effective for
indoor animals. Therefore, increasing the proportion of intensive livestock production is
conducive to better manure management (Hristov et al., 2011). Secondly, the ratios of $NH_3$
emission reductions mentioned above were multiplied by $NH_3$ emission factors in three
phases of manure management: 50% reduction at in-house handling, 60% (average value
of 50% and 70%) reduction at storage and 50% reduction at land application. With these
measures, we estimate that the $NH_3$ emission factors for the livestock in China could be
comparable to those in Europe and the USA (shown in Table S1). Meanwhile, the $NH_3$
emission model predicted that the livestock $NH_3$ emissions were reduced by 60% (from 57
to 23 kiloton), causing approximately 40% reduction in total $NH_3$ emissions. Spatially,
$NH_3$ emissions decreased significantly in Hebei, Henan and Shandong, where the livestock
$NH_3$ emissions accounted for a large proportion of the total (shown in Figure S3).
**3.2. Simulations of $NO_3^-$ reduction due to $NH_3$ emission controls**
In the ISORROPIA-II simulation, 40% reduction of TA was used to reflect the effects
of reducing $NH_3$ emissions by 40%. In this haze event (from 6 to 10 December, 2015), the
mean concentration of particulate $NO_3^-$ decreased from 40.8 to 25.7 μg/m$^3$ (a 37%
reduction). In addition, the peak hourly concentration of $NO_3^-$ decreased from 81.9 to 30.7
μg/m$^3$ (a 63% reduction) (shown in Figure 1). The fundamental thermodynamic processes
of TA reductions on decreasing particulate $NO_3^-$ are explained below. Firstly, we found that
$NH_3$ was quite available to react with $HNO_3$ in the thermodynamic equilibrium system,
because $NH_3$ was 6.6 ± 3.8 μg/m$^3$ while $HNO_3$ was only 0.4 ± 1.1 μg/m$^3$. Secondly, almost
all of particulate $NO_3^-$ condensed into aerosol phase (the mass ratio of particulate $NO_3^-$ to



TN was 99.2 ± 1.9%) under such low temperature conditions (276.5 ± 1.4 K). Thirdly, the
$NH_3$-$HNO_3$ partial pressure production ($K_p$) was as low as about 0.1 $ppb^2$ (calculated from
ISORROPIA-II outputs, depending not only on temperature and RH but also sulfate
concentration). The value of $K_P$ would remain constant, if the temperature, RH and sulfate
concentration remained unchanged. In general, $NH_4NO_3$ was not easy to volatilize into gas
phase under these circumstances.

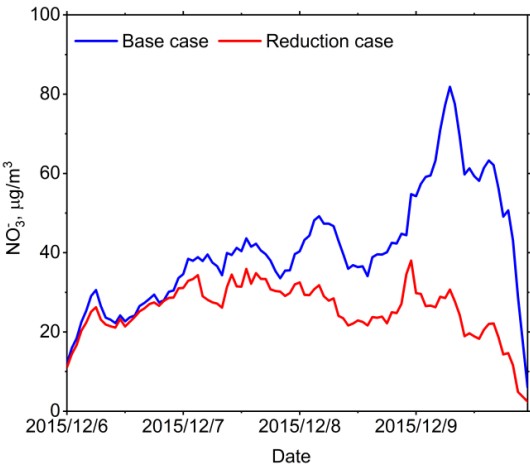


**Figure 1.** A comparison of particulate nitrate ($NO_3^-$) between the base (blue line) and
emission reductions cases (red line) simulated by the ISORROPIA-II model in this severe
haze episode.

When TA was reduced by 40%, the average mass concentration of gaseous $NH_3$
decreased from 6.6 to 0.01 $\mu g/m^3$ (from 8.8 ppb to 0.05 ppb). In order to keep the value of
$K_P$ constant in the thermodynamic equilibrium state, the reductions of $NH_3$ increased $HNO_3$,
which shifted the particulate $NO_3^-$ partitioning toward the gas phase. Hence, when $NH_3$ in
gas phase was almost completely depleted, $HNO_3$ increased from 0.4 to 15.5 $\mu g/m^3$ (from
0.1 ppb to 5.6 ppb), leading to a reduction of particulate $NO_3^-$ from 40.8 to 25.7 $\mu g/m^3$ (a
37% reduction). Meanwhile, $NH_4^+$ also decreased from 27.9 to 20.6 $\mu g/m^3$ and there was
almost no change in sulfate level (decreased from 39.7 to 39.3 $\mu g/m^3$), with only trace
amount of $NH_4HSO_4$ produced. This indicated that the reduction of particulate $NH_4^+$ and
$NO_3^-$ was mainly due to the reduction of $NH_4NO_3$.

The above process could also be explained by the interactions between aerosol acidity
and gas-particle partitioning of $HNO_3$. Guo et al. (2018) used the S curves to demonstrate
the relationship between aerosol pH and $HNO_3$ partitioning in the United States, Europe
and China. The results showed that when TA was reduced to a certain extent, aerosol pH
began to decline, prompting the particulate $NO_3^-$ volatilizing into gas phase. However,
using aerosol pH as an indicator of the sensitivity of particulate $NO_3^-$ to TA has some
limitations. On the one hand, it may not be suitable for low temperature or low relative
humidity conditions. Because under these conditions, the estimation of aerosol pH would





become inaccurate (Fountoukis et al., 2009). On the other hand, recent studies showed
many factors could cause bias in aerosol pH prediction. For instance, Vasilakos et al. (2018)
found that non-volatile cations ($K^+$, $Na^+$, $Ca^{2+}$ and $Mg^{2+}$) in the fine mode could cause bias
in the aerosol pH and $HNO_3$ partitioning prediction. Silvern et al. (2017) found that
particles coated by organic material might retard the uptake of $NH_3$, which may also cause
bias in modeling aerosol acidity and $HNO_3$ partitioning. In general, studying the process
of aerosol acidity affecting particulate $NO_3^-$ formation still requires more work to do,
especially sensitivity tests, to unravel the potential effects of other factors.
We also conducted WRF-Chem simulations to quantify the impacts of $NH_3$ emission
controls on particulate $NO_3^-$ regionally. A 60% reduction in livestock $NH_3$ emissions was
used as an emission reductions scheme and Figure 2 shows the spatial distribution of
particulate $NO_3^-$ under the base case and the emission reductions case. The spatial
distribution of particulate $NO_3^-$ was mainly concentrated in most parts of Henan (HN) and
Hebei (HB), with the average concentration over 30 $\mu g/m^3$ (included in the blue box shown
in Figure 2a). The highest particulate $NO_3^-$ concentrations, more than 60 $\mu g/m^3$, were
mainly located in central south of Hebei and northern Henan. In the emission reductions
case, the mean concentration of particulate $NO_3^-$ decreased from 34.2 to 20.7 $\mu g/m^3$ (a 40%
reduction) in the range of the blue box. In addition, the sulfate concentration slightly
changed from 28.1 to 24.3 $\mu g/m^3$, and $PM_{2.5}$ concentration dropped from 161.7 to 139.3
$\mu g/m^3$. The largest reductions in particulate $NO_3^-$ were mainly located in the central north
of Henan and central Hebei, where the percentage reduction was generally more than 60%
(shown in Figure 2b). In these regions, severe haze events occurred frequently due to their
large emissions of air pollutants, including $NH_3$ (Wang et al., 2014;Zhao et al., 2017). The
contrast of figure 2a and 2b shows that particulate $NO_3^-$ had been effectively reduced,
especially in high concentration areas. The reason is explained in Sect. 3.3.

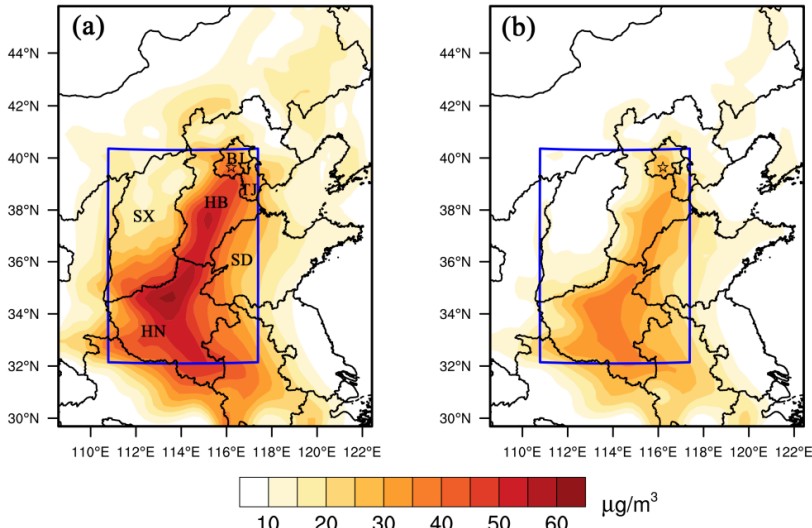


**Figure 2. (a)** Spatial distribution of particulate $NO_3^-$ concentrations in northern China
predicted by WRF-Chem from 6 to 10 December, 2015, for **(a)** the base case, and **(b)** the



emission reductions case. The scope of this study focuses on the blue box, including Beijing
(BJ), Tianjin (TJ), Hebei (HB), Shanxi (SX), Shangdong (SD) and Henan (HN).

### 3.3 The particulate $NO_3^-$ reduction efficiency during the wintertime

The sensitivity of particulate $NO_3^-$ to $NH_3$ is often determined by the availability of
ambient $NH_3$, which can be represented by the observable indicator (Seinfeld and Pandis,
2006). In this study, we use the observed molar ratio (R) of TA to the sum of sulfate, total
chlorine and TN minus $Na^+$, $K^+$, $Ca^{2+}$ and $Mg^{2+}$ to represent the availability of ambient
$NH_3$ and predict the sensitivity of the particulate $NO_3^-$ to changes in TN and TA.
$$R = \frac{TA}{2SO_4^{2-}+NO_3^-+HNO_3(g)+Cl^-+HCl(g)-2Ca^{2+}-Na^+-K^+-2Mg^{2+}} \quad (1)$$
The accuracy of R was examined by constructing the isopleths of particulate $NO_3^-$
concentrations as a function of TN and TA (shown in Figure 3). The $NO_3^-$ concentration
was constructed by varying the input concentrations of TA and TN from 0 to 200 μg/m³ in
increments of 10 μg/m³ independently in ISORROPIA-II, while using the observed average
value for the other components. Over a range of temperatures (273–283 K) and RHs (30–
90%), the dashed line of R = 1 divides each isopleth into two regions with tiny bias, which
indicates that R can be used to qualitatively predict the response of the particulate $NO_3^-$ to
changes in concentrations of TN and TA.

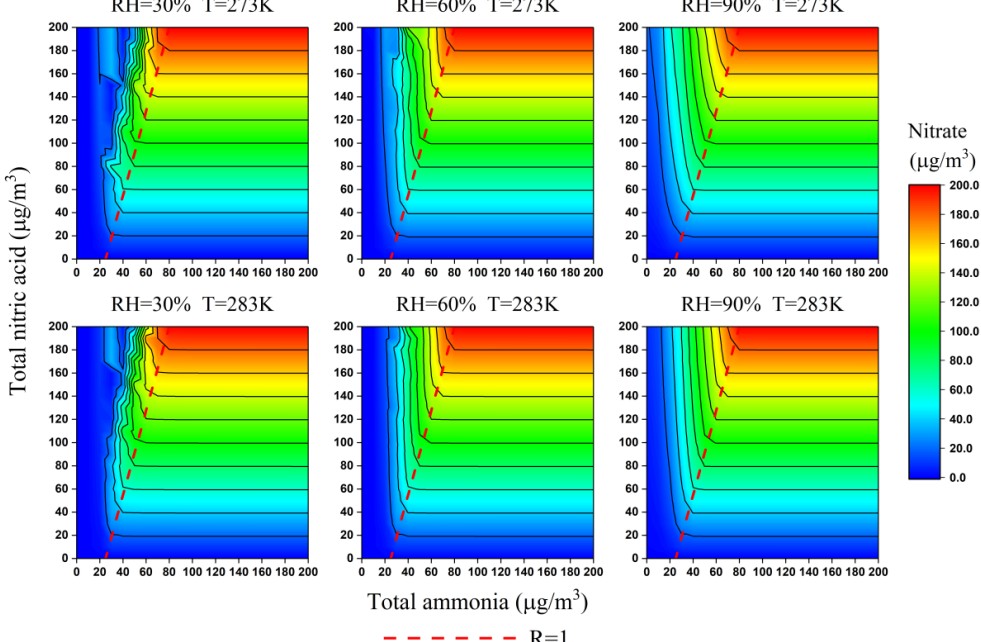


**Figure 3.** Isopleths of the particulate $NO_3^-$ concentration (μg/m³) as a function of TN
and TA under average severe haze conditions in winter. The concentration of $SO_4^{2-}$, $Cl^-$, $K^+$,





$Ca^{2+}$, $Na^+$, and $Mg^{2+}$ was 60.2, 9.3, 0.56, 0.04, 0.75, and 0.03 μg/m$^3$, respectively. Values
are averages from all severe hazes during the observation period.
In the right side of the dashed line (R > 1), particulate $NO_3^-$ formation is $HNO_3$-limited.
The $NH_3$ is surplus and almost all particulate $NO_3^-$ exists in the aerosol phase. The TA
reductions mainly reduce $NH_3$, with negligible effects on particulate $NO_3^-$. By contrast,
particulate $NO_3^-$ formation is $NH_3$-limited in the left of the dashed line (R < 1). There is
less $NH_3$ present in the gas phase, and TA reductions could reduce particulate $NO_3^-$
efficiently. For example, when the concentrations of TN and TA are 100 and 50 μg/m$^3$ (RH
= 60 % and T =273 K), the concentration of particulate $NO_3^-$ is about 100 μg/m$^3$ and the
value of R is close to one (typical observational values during the severe haze in this study).
In such cases, if TA were reduced by 50% to 25 μg/m$^3$, the particulate $NO_3^-$ would be
significantly reduced from 100 to 20 μg/m$^3$, an 80% reduction.
Under the typical winter conditions in northern China, the value of R was generally
greater than one and gradually declining with the increase in SNA concentrations (shown
in Figure 4a). When the concentration of SNA is greater than 150 μg/m$^3$, the values of R
become close to and frequently lower than one. This indicated that particulate $NO_3^-$
formation would easily become $NH_3$-limited under severe haze conditions when $NH_3$
emissions were reduced. In general, particulate $NO_3^-$ will be reduced effectively by a 40%
reduction of $NH_3$ emissions in the condition that the value of R is less than 1.4 (shown in
Figure S4). This situation accounts for 68.1% of the entire December (shown in Figure 4b).
It should also be noted that the particulate $NO_3^-$ can be insensitive to a 40% reduction in
$NH_3$ emissions when the value of R is greater than 1.4 (shown in Figure S4). This situation
mainly occurs in relatively clean days (the concentration of SNA is less than 75 μg/m$^3$),
accounting for only 31.9% of the entire December (shown in Figure 4a and 4b). Overall,
reducing 40% of $NH_3$ emissions could effectively reduce the levels of particulate $NO_3^-$
under typical winter haze conditions in northern China.

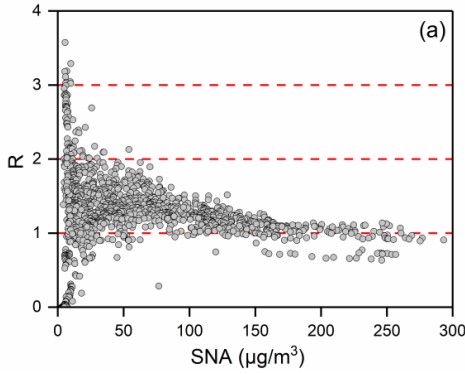
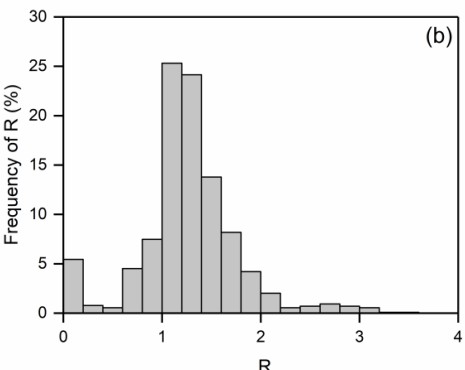


**Figure 4.** (a) The observed molar ratio (R) and the concentrations of SNA in PKUERS
in December 2015 and December 2016. (b) The frequency of R during the same period.




## 4 Discussions

Improved manure management could reduce 40% of $NH_3$ emissions in winter of northern China. For a 40% reduction of TA in the atmosphere, ISORROPIA-II predicts that particulate $NO_3^-$ could be reduced by 37% for the haze event (from 40.8 to 25.7 $\mu g/m^3$). When $NH_3$ emissions are reduced by 40%, the WRF-Chem simulation shows that particulate $NO_3^-$ concentration could be reduced effectively throughout the whole region (the mean concentration of particulate $NO_3^-$ was reduced from 34.2 to 20.7 $\mu g/m^3$, a 40% reduction), especially in the area with high particulate $NO_3^-$ concentration (more than 60%). The molar ratio (R) of winter observational data in northern China shows that as the concentration of inorganic salts increases, the excess $NH_3$ in the atmosphere is decreasing, and particulate $NO_3^-$ becomes more sensitive to $NH_3$ emission reductions. In general, controlling livestock $NH_3$ emissions could be an effective measure to reduce $NH_3$ levels and limit particulate $NO_3^-$ formation.

The observed R provides a simple method to rapidly estimate the efficiency of $NH_3$ emission reductions on the particulate $NO_3^-$ reductions, which can avoid the shortage of the air quality model, especially the uncertain estimates of meteorology. However, it also has some limitations, such as requiring accurate measurements of water-soluble ions and gaseous components. In addition, the accuracy of R needs to be examined in more detail for specific pollution and meteorological conditions. Therefore, the observed indicator and air quality models should be used in a complementary way to assess the effectiveness of $NH_3$ emission controls strategies.

$NO_X$ emission controls could be a more direct and effective way to reduce the particulate $NO_3^-$ than $NH_3$ emission reductions. However, in northern China, the target of $NO_X$ emission reductions is only about 25% in the 13th Five-Year Plan (2016-2020). Furthermore, the previous study has shown that $NO_X$ emission reductions in the U.S could be much costly to control $PM_{2.5}$ than reductions in $NH_3$ emission (Pinder et al., 2007). Due to the dominance of extensive livestock farming and the lack of emission controls policies, $NH_3$ emission reductions in China may be easier and cost less than the U.S. Therefore, in order to control $PM_{2.5}$ pollution more effectively in northern China, $NH_3$ emission controls are urgently needed. In addition, a comprehensive cost-effectiveness analysis of $NH_3$ emission control strategies in China is also much needed in the future.

It should be noted that the $NH_3$ emission reductions have plenty of significant environmental implications. On the one hand, it could increase the particle acidity to increase the solubility of transition metals in aerosol phase, which is related to aerosol oxidative stress and toxicity (Fang et al., 2017;Longo et al., 2016). Meanwhile, metal mobility could affect photosynthesis productivity (Duce and Tindale, 1991;Li et al., 2017c) and oxygen levels in the ocean (Ito et al., 2016) by changing nutrient distributions. Furthermore, particle strong acidity has also been directly linked to adverse respiratory effects (Schlesinger, 2007;Ward et al., 2002). On the other hand, anthropogenic $NH_3$ emission controls can decrease excess reactive nitrogen inputs to earth ecosystems, which could alleviate many adverse ecological effects including soil acidification, plant biodiversity reduction, and eutrophication (Bouwman et al., 2002;Stevens et al., 2004;Bowman et al., 2008). All these environmental impacts in China require further research.





## Acknowledgement

This study was supported by National Key R&D Program of China (2016YFC0201505)
and National Natural Science Foundation of China (NSFC) (91644212, 41675142,
21625701 and 41571130033). The data used in this study are available from the
corresponding author. The authors declare no competing interests.

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
