# Peer review of "High efficiency of livestock ammonia emission controls on alleviating particulate nitrate during a severe winter haze episode in northern China"

_Atmospheric Chemistry and Physics, 2018_

## Referee Comment (RC1) · Anonymous Referee #1 · 29 Oct 2018

Xu et al. applied a model analysis, and found "High efficiency of livestock ammonia emission controls on alleviating particulate nitrate during a severe winter haze episode in northern China". The research topic is of extreme importance for adding scientific knowledge and supporting policy-makers on ammonia controls from livestock sector. The most important finding is that 40% of ammonia emission mitigation could lead to almost the same reduction in particulate nitrate in the North China Plain in winter season. This finding (based on real-time IGAC measurements and atmospheric modeling) provides strong evidence of the importance of livestock $NH_3$ mitigation (combined with

[Figure]

NOx and SO2 emission reductions) in improving air quality in this intensive agricultural and industrial region. Nevertheless, several statements & discussions are needed to be clarified in this manuscript. I suggest the manuscript to be published in ACP after proper revisions as below.

Major comments 1. General. While this paper could be useful as a theoretic support of ammonia emission controls on alleviating particulate matters, however, the authors should express their new findings (e.g. the detailed analysis of the equilibrium between . . .) clearly in the revision. Because it is not surprising that a reduction in NH3 emission alleviates particulate matter (e.g. PM2.5) pollution (see Wu Y. et al., 2016; Wu S.-Y. et al., 2008; Backes et al., 2016; Pinder et al., 2007). Refs mentioned: Y. Wu, B. Gu, J. W. Erisman, S. Reis, Y. Fang, X. Lu, X. Zhang, PM2.5 pollution is substantially affected by ammonia emissions in China. Environmental Pollution 218, 86-94 (2016). S.-Y. Wu, J.-L. Hu, Y. Zhang, V. P. Aneja, Modeling atmospheric transport and fate of ammonia in Carolinaâ˘ATPart II: Effect of ammonia emissions on fine particulate matter formation. Atmospheric Environment 42, 3437-3451 (2008). A. M. Backes, A. Aulinger, J. Bieser, V. Matthias, M. Quante, Ammonia emissions in Europe, part II: How ammonia emission abatement strategies affect secondary aerosols. Atmospheric Environment 126, 153-161 (2016). R. W. Pinder, P. J. Adams, S. N. Pandis, Ammonia Emission Controls as a Cost-Effective Strategy for Reducing Atmospheric Particulate Matter in the Eastern United States. Environmental Science & Technology 41, 380-386 (2007).

2. Methodology. The use of WRF model did not reproduce the temporal variations of inorganic aerosol components in this haze event (Figure S2 in the supporting information). As shown in Fig. S2, the correlation between the observations and simulations was relatively low, but the authors did not show this value deliberately. Due to such low accuracy of the WRF to simulate the inorganic aerosol components, how can the authors draw such strong conclusions based an unconvincing simulations? I suggest the authors validate their simulations using the observations, make some improvements

of the simulation ability, and discuss the potential biases of the simulations; or alternatively, discuss the uncertainties of the simulation results in the discussions section. This is important because it's the fundamental base for your conclusions.

3. Form and structure. There are well known heterogeneities in the NH3 emission datasets that would need to be discussed in detail (refer to Zhang et al, 2018, Agricultural ammonia emissions in China reconciling bottom-up and top-down estimates. Atmospheric Chemistry and Physics, 18: 339-355). In the authors' estimates, the livestock NH3 emission is in general lower than 1.8 kg NH3 ha-1 (180 kg NH3 km-2) (Fig. S3). It is such low livestock NH3 emission in northern China in December. Is it right? And why such low livestock NH3 emission have so big impact on particular matters? I wonder if the unit of NH3 emission is kg NH3 ha-1 month-1 ?

The authors had good measurements dataset of the inorganic aerosol components during in December 2015 and December 2016. Unfortunately, it is very surprising that the authors made a conclusion based the simulation data rather than their measurements. If the authors want to make a strong conclusion that livestock ammonia emission controls on alleviating particulate nitrate during a severe winter haze, they should first show what they has gained from the two time periods of December 2015 and December 2016 regarding the measurements of inorganic aerosol components as well as their estimates of livestock NH3 emissions? Again, the simulation results are unacceptable for inorganic aerosol components from the two time periods of December 2015 and December 2016. The conclusion should be based on their measurements work. At least, their simulations should be finely validated with their observations.

Specific comments Introduction 1. line 66-71 these review introductions are very lacking, and numerous studies on this topic have been ignored by the authors, which I have given several of them above. It is impossible for the reader to judge what the merits are of the current paper without ploughing through the recent literature, which as pointed out before is not properly reviewed.

Methods 1. Line 83: the authors said the measurements were conducted in December 2015 and December 2016. Why are the results of December 2016 not shown in the paper, and why the validation was only performed in December 2015 (Fig. S2)?. 2. Line 86: HCl (rather than HCI). 3. Line 96-110: The validation of the livestock NH3 emission products should be described in detail.

Results 1. Line 61: "On the one hand, the proportion of intensive livestock husbandry in China is only about 40%, far lower than that of developed countries". What's the proportion of intensive livestock husbandry in developed countries (90% or 100%)? At least, a reference should be given here. 2. Lines 165-170: these statements are very biased since their study timespan concerned the winter time (December), while the N application commonly occurred in spring or summer. The authors should focus on the timespan of their study, and avoid overstatements of their findings. 3. Lines 171-197: Again these statements are overstated. Actually, the authors just make a very subjective reduction in livestock NH3 emissions, and then drive the WRF model using the reduced livestock NH3 emission. 4. Lines 199-200: In the ISORROPIA-II simulation, 40% reduction of TA was used to reflect the effects of reducing NH3 emissions by 40%. This process is also very subjective and has no explanation at all why the authors adopted this value. At least the author should give reference to support this process. In fact, there are numerous subjective descriptions in the main text, and it's hard to specify all of them and prove them validate.

Discussions 1. Lines 319-336: All these were already shown in results part, but were again repeated in the discussions. I suggest the authors re-organize the discussions sector in order to summarize their results completely, also for better comparison to some latest references.

Please also note the supplement to this comment:
https://www.atmos-chem-phys-discuss.net/acp-2018-896/acp-2018-896-RC1-supplement.pdf

---

## Referee Comment (RC2) · Anonymous Referee #2 · 12 Nov 2018

This is a straightforward and concise analysis of the sensitivity of particle nitrate loadings to winter haze episodes in Northern China. It addresses an important question – how to effectively reduce particle loadings under conditions of very bad air quality. The authors argue that because a significant proportion of Northern China's NH3 emissions during the winter come from livestock, and because current agricultural practices lead to high emissions which could be reduced relatively easily (by 60% through adopting practices more common in Europe and the U.S.), that reducing total NH3 emissions by 40% in the winter in achievable. Based on this argument, the paper pursues two

complimentary approaches to testing the sensitivity of particle nitrate to reductions of NH3. In the first, they use thermodynamic modelling of a comprehensive observational dataset obtained from measurements at a single site. While the modelling is not perfect, especially in terms of its performance for gas phase species, the authors make the case that the model results are robust for the particle phase and thus reliable for predictions when particle mass loadings are high. By applying a consistent 40% reduction to total ammonia (TA) mass loading, they find a significant reduction in particle nitrate that grows in absolute and relative terms over the course of a 4-day haze event.

To take a more holistic approach, the authors also perform WRF-Chem simulations over a domain centred on Northern China, performing a base case run and one in which NH3 emissions from livestock were decreased by 60%. The authors make the argument for this more sophisticated approach in part because the non-linear relationship between ammonia and nitrate could change lifetime of nitrate. The authors miss an opportunity to test whether this is true under their conditions. I would encourage them to calculate the change in total nitrate (TN) burden (and/or lifetime) as a result of changing the NH3 emissions. They should also calculate the change in TA burden (and/or lifetime) to determine in a reduction in concentration of 40% is the result. Because the WRF-Chem simulations do a relatively poor job in representing TA at the observation site, confidence in the model predictions is undermined. In part 3.3, the authors use the metric of molar ratio (R) to explain under what conditions particle nitrate is sensitive to reductions in TA vs TN. It would be useful if they could place their model simulation results in the context of this framework. If the model is biased in TA (or TN) but occupies a relatively 'flat' part of the isopleth diagram, then its predictions could still be robust. But if biases in the model lead to changes in R near 1, then the predictions may not be as reliable.

Specific comments

In the abstract and throughout the text, the authors consistently focus on the reduction in particle nitrate loading that results from reductions in NH3 emissions, but particle

ammonium levels also change. While the absolute change in mass loading of ammonium will be less than nitrate due to its lower molecular weight, it would still be worth it in a couple of instances to calculate and report the total reduction in PM2.5 mass from nitrate AND ammonium.

Section 2.2 More information should be provided about the inventory. Over what geographic area are the emissions quoted for? 'North China' is referred to several times, but it would be useful to be more specific. Is the region under study the totality of the six provinces shown in Figure 2, or just the area within the blue box in Figure 2? Or the domain in Figure S3? Also, is the inventory used in this work archived and available for public access?

Figure 2 – I suggest adding a third panel that shows either the absolute difference between the two model runs or the percent decrease. It would be useful to see the spatial pattern of the change in nitrate.

Figure S3 – Is it kg of N in NH3 or kg of NH3 itself?

---

## Author Comment (AC1) · 4 Jan 2019

Our point-by-point responses are provided below. The referees' comments are italicized.

**Response to Referee #1**

*Referee: Xu et al. applied a model analysis, and found "High efficiency of livestock ammonia emission controls on alleviating particulate nitrate during a severe winter haze episode in northern China". The research topic is of extreme importance for adding scientific knowledge and supporting policy-makers on ammonia controls from livestock sector. This finding (based on real-time IGAC measurements and atmospheric modeling) provides strong evidence of the importance of livestock $NH_3$ mitigation (combined with NOx and $SO_2$ emission reductions) in improving air quality in this intensive agricultural and industrial region. Nevertheless, several statements & discussions are needed to be clarified in this manuscript. I suggest the manuscript to be published in ACP after proper revisions as below.*

**Response:** We would like to thank the referrer for your detailed and constructive comments. Please see our point-by-point reply below.

*Referee: 1. General.*

*While this paper could be useful as a theoretic support of ammonia emission controls on alleviating particulate matters, however, the authors should express their new findings (e.g. the detailed analysis of the equilibrium between ...) clearly in the revision. Because it is not surprising that a reduction in $NH_3$ emission alleviates particulate matter (e.g. $PM_{2.5}$) pollution (see Wu Y. et al., 2016; Wu S.-Y. et 23 al., 2008; Backes et al., 2016; Pinder et al., 2007).*

**Response:** Accepted. There are three new findings in our study. 1. During severe winter haze episodes, the particulate $NO_3^-$ formation is $NH_3$-limited, resulting in its high sensitivity to $NH_3$ emission reductions. 2. Livestock $NH_3$ emission controls is a very efficient way to alleviate particulate $NO_3^-$ pollution during severe winter hazes. 3. Improved manure management in livestock husbandry could effectively reduce total $NH_3$ emissions by 40% (from 100 kiloton to 60 kiloton) in winter of northern China, which would lead to a reduction of particulate $NO_3^-$ by about 40% (averagely from 40.8 to 25.7 $\mu g/m^3$) during severe haze conditions. As you suggested, we reworded in the revised manuscript.

**Revision:** (Page 12, Line 381-388) "In this study, we found that during severe winter haze episodes, the particulate $NO_3^-$ formation is $NH_3$-limited, resulting in its high sensitivity to $NH_3$ emission reductions. Meanwhile, livestock $NH_3$ emission controls is a very efficient way to alleviate particulate $NO_3^-$ pollution during severe winter hazes. The estimations showed that the improvements in manure management of livestock husbandry could effectively reduce total $NH_3$ emissions by 40% (from 100 kiloton to 60 kiloton) in winter of northern China. It would lead to a reduction of

particulate $NO_3^-$ by about 40% (averagely from 40.8 to 25.7 μg/m$^3$) during severe haze conditions."

Referee: 2. Methodology.

***The use of WRF model did not reproduce the temporal variations of inorganic aerosol components in this haze event (Figure S2 in the supporting information). As shown in Fig. S2, the correlation between the observations and simulations was relatively low, but the authors did not show this value deliberately. Due to such low accuracy of the WRF to simulate the inorganic aerosol components, how can the authors draw such strong conclusions based an unconvincing simulations? I suggest the authors validate their simulations using the observations, make some improvements of the simulation ability, and discuss the potential biases of the simulations; or alternatively, discuss the uncertainties of the simulation results in the discussions section. This is important because it's the fundamental base for your conclusions.***

**Response:** Accepted. As you suggested, we improved our model performance and added discussions of the simulation biases and their impacts in Section 2.2 and 3.3, respectively. Averagely, the observed and simulated $NO_3^-$, $NH_4^+$, $SO_4^{2-}$ and TA are, respectively: (1) $NO_3^-$, 39.8 ± 14.7 μg/m$^3$ versus 39.1 ± 15.6 μg/m$^3$; (2) $NH_4^+$, 27.7 ± 8.6 μg/m$^3$ versus 26.5 ± 11.7 μg/m$^3$; (3) $SO_4^{2-}$, 42.4 ± 16.0 μg/m$^3$ versus 39.7 ± 20.8 μg/m$^3$ and (4) TA, 34.6 ± 8.5 μg/m$^3$ versus 32.1 ± 11.0 μg/m$^3$. The MB of these four species are -0.7, -1.2, -2.7 and -2.5 μg/m$^3$, respectively. Simulated particulate $NO_3^-$, $NH_4^+$, $SO_4^{2-}$ and TA approximately agreed with the measurements.

In fact, we used 1-hr resolution measurements to compare with the simulations. The severe hazes often happened in stagnant conditions, in which the turbulent diffusion is weak and the winds almost keep calm. In this situation, it is very difficult for chemical transport models (like WRF-Chem) to describe the local atmospheric stability or diffusion processes very well (Steeneveld et al., 2006;Steeneveld, 2014). Moreover, the uncertainty in emissions could not be neglected. These factors make it difficult for chemical transport models to reproduce the temporal variations of inorganic aerosol components very well at hourly resolution (Li et al., 2016).

The simulation biases may affect the simulation of particulate $NO_3^-$ reductions efficiency. Based on our results in Section 3.3, particulate $NO_3^-$ reduction efficiency is determined by the availability of ambient $NH_3$ (represented as R in this study). Correspondingly, the influence of simulation biases on particulate $NO_3^-$ reduction efficiency simulation mainly depends on the simulation biases of R. During the simulation case, the average simulated value of R is 1.3, which is equivalent to the observed value (1.3). Since WRF-Chem has a good estimation of the availability of ambient $NH_3$, its estimation of the efficiency of particulate $NO_3^-$ reductions

is reliable. Therefore, the conclusions drawn in Sect 3.2 are reliable.

**Revision:** (Page 5, Line 185-193) "The performance of WRF-Chem is evaluated by comparing measured and simulated $NO_3^-$, $NH_4^+$, $SO_4^{2-}$ and TA. Specifically, the observed and simulated values are, respectively: (1) $NO_3^-$, $39.8 \pm 14.7$ $\mu g/m^3$ versus $39.1 \pm 15.6$ $\mu g/m^3$; (2) $NH_4^+$, $27.7 \pm 8.6$ $\mu g/m^3$ versus $26.5 \pm 11.7$ $\mu g/m^3$; (3) $SO_4^{2-}$, $42.4 \pm 16.0$ $\mu g/m^3$ versus $39.7 \pm 20.8$ $\mu g/m^3$ and (4) TA, $34.6 \pm 8.5$ $\mu g/m^3$ versus $32.1 \pm 11.0$ $\mu g/m^3$. The MB of these four species are -0.7, -1.2, -2.7 and -2.5 $\mu g/m^3$, respectively. Simulated particulate $NO_3^-$, $NH_4^+$, $SO_4^{2-}$ and TA approximately agreed with the measurements (Figure S2). There are still some simulation biases that may affect the simulation of particulate $NO_3^-$ reductions efficiency. This is discussed in detail in Sect 3.3."

(Page 12, Line 365-369) "Based on the above analysis, the influence of WRF-Chem simulation biases on particulate $NO_3^-$ reduction efficiency simulation mainly depends on the simulation bias of R. During the simulation case, the average simulated value of R is 1.3, which is equivalent to the observed value (1.3). Since WRF-Chem has a good estimation of the availability of ambient $NH_3$, its estimation of the efficiency of particulate $NO_3^-$ reductions is reliable."

Referee: 3. Form and structure.

*There are well known heterogeneities in the $NH_3$ emission datasets that would need to be discussed in detail (refer to Zhang et al, 2018, Agricultural ammonia emissions in China reconciling bottom-up and top-down estimates. Atmospheric Chemistry and Physics, 18: 339-355).*

**Response:** Accepted. As you suggested, we added more descriptions about the heterogeneities in the $NH_3$ emission datasets in Sect 2.2.

**Revision:** (Page 4, 127-141) "Another method for estimating $NH_3$ emissions is the inverse modeling method, which provides top-down emission estimates through optimizing comparisons of model simulations with measurements. For example, Paulot et al. (2014) used the adjoint of a global chemical transport model (GEOS-Chem) and data of $NH_4^+$ wet deposition fluxes to optimize $NH_3$ emissions estimation in China. Zhang et al. (2018a) applied TES satellite observations of $NH_3$ column concentration and GEOS-Chem to provide top-down constraints on $NH_3$ emissions in China. Their estimates are 10.2 Tg $a^{-1}$ and 11.7 Tg $a^{-1}$ respectively, which are close to our results (9.8 Tg $a^{-1}$) (Paulot et al., 2014;Zhang et al., 2018a). The accuracy of this method relies on many factors, such as the accuracy of initial conditions, the emission inventories, meteorological inputs, reaction rate constants, and deposition parameters in the chemical transport model. Errors of these parameters could cause biases in the top-down estimation of $NH_3$ emissions. In addition, measurements of $NH_3$ or $NH_4^+$ used in this

method, including surface and satellite date, are usually sparse in spatial coverage and have uncertainties, which will also affect the estimation of $NH_3$ emissions."

*In the authors' estimates, the livestock $NH_3$ emission is in general lower than 1.8 kg $NH_3$ ha-1 (180 kg $NH_3$ $km^{-2}$) (Fig. S3). It is such low livestock $NH_3$ emission in northern China in December. Is it right? And why such low livestock $NH_3$ emission have so big impact on particular matters? I wonder if the unit of $NH_3$ emission is kg $NH_3$ $ha^{-1}$ $month^{-1}$?*

**Response:** Yes, the correct unit is kg $NH_3$ $km^{-2}$ $month^{-1}$. Figure S4 has been revised.

*The authors had good measurements dataset of the inorganic aerosol components during in December 2015 and December 2016. Unfortunately, it is very surprising that the authors made a conclusion based the simulation data rather than their measurements. If the authors want to make a strong conclusion that livestock ammonia emission controls on alleviating particulate nitrate during a severe winter haze, they should first show what they has gained from the two time periods of December 2015 and December 2016 **regarding the measurements of inorganic aerosol components as well as their estimates of livestock $NH_3$ emissions**? Again, the simulation results are unacceptable for inorganic aerosol components from the two time periods of December 2015 and December 2016. The conclusion should be based on their measurements work. At least, their simulations should be finely validated with their observations.*

**Response:** Firstly, in fact, our conclusions are mainly based on measurements. In the ISORROPIA-II simulation, the input data are all the observation data and we show the comparison between observed and simulated particulate $NO_3^-$ after TA reductions. In addition, the analysis of the availability of ambient $NH_3$ in Section 3.3 is also based entirely on observations. In the WRF-Chem simulation, because we needed to show the particulate $NO_3^-$ reductions regionally, we calculated the change of simulated value of particulate $NO_3^-$ before and after $NH_3$ emission reductions.

Secondly, our observations and $NH_3$ emission inventory have been described in detail in section 2.1 and 2.2. The importance of particulate $NO_3^-$ in SNA and the dominant role of livestock in $NH_3$ emissions are pointed out. Furthermore, from lines 249 to 259, we made a conclusion that the richness of $NH_3$ leads to the stability of $NH_4NO_3$ in the atmosphere by calculating the $NH_3$-$HNO_3$ partial pressure production (Kp) and analyzing the phase state and composition of pollutants. This conclusion directly linked high $NH_3$ emissions to high particulate $NO_3^-$ concentrations, which is also based entirely on observations.

Thirdly, as you suggested, we discussed the simulation biases and their

impacts in Section 2.2 and 3.3, respectively. See the previous reply for details.

Specific comments:

**Introduction**

1. *line 66-71 these review introductions are very lacking, and numerous studies on this topic have been ignored by the authors, which I have given several of them above. It is impossible for the reader to judge what the merits are of the current paper without ploughing through the recent literature, which as pointed out before is not properly reviewed.*

Response: Accepted. As you suggested, we added more review introductions to highlight the importance and innovation of our research.

Revision: (Page 2-3, Line 48-88) "In northern China (including Beijing, Tianjin, Hebei, Shandong, Shanxi and Henan), severe haze pollution events occur frequently during wintertime, with the concentration of $PM_{2.5}$ (particles with an aerodynamic diameter less than 2.5 μm) reaching hundreds of micrograms per cubic meter and SIA (secondary inorganic aerosol) accounting for more than 50% of $PM_{2.5}$ (Zheng et al., 2016;Tan et al., 2018). To mitigate fine particle pollution, the Chinese government has been taking strong measures to control $SO_2$ emissions (http://www.gov.cn/zwgk/2011-12/20/content_2024895.htm). Since 2007, $SO_2$ emissions have been reduced by 75% in China (Li et al., 2017). Consequently, the particulate sulfate concentration have also been declining continuously in the past decade (Geng et al., 2017).

Although $NO_x$ emissions in 48 Chinese cities decreased by 21% from 2011 to 2015 (Liu et al., 2017a), unfortunately, no obvious decreasing trend for particulate $NO_3^-$ had been observed in northern China during recent years (Zhang et al., 2015). In October 2015, a severe haze episode was reported in North China Plain (NCP), with the hourly peak concentration of particulate $NO_3^-$ exceeding 70 μg/m$^3$ (Zhang et al., 2018b). Even in November 2018, during a heavy haze episode in northern China, the hourly peak concentration of $PM_{2.5}$ still exceeded 289 μg/m$^3$, of which particulate $NO_3^-$ accounted for 30% (http://www.mee.gov.cn/xxgk2018/xxgk/xxgk15/201811/t20181116_674022.html).

Another way to alleviate the particulate $NO_3^-$ pollution is to control $NH_3$ emissions. Previous studies were performed to demonstrate the necessity of $NH_3$ emissions abatement in reducing $PM_{2.5}$ concentrations in the United States (Pinder et al., 2007;Tsimpidi et al., 2007;Pinder et al., 2008;Wu et al., 2016) and Europe (de Meij et al., 2009;Bessagnet et al., 2014;Backes et al., 2016). Recently, a feature article pointed out that $NH_3$ could be key to

limiting particulate pollution (Plautz, 2018). In contrast with low particulate matter pollution levels in the United States and Europe, what we are facing in northern China is the extremely high particulate $NO_3^-$ pollution especially happened in severe winter haze events.

Although Fu et al. (2017) proposed that the $NH_3$ emission controls are urgently required in China, the effectiveness of $NH_3$ emissions mitigation to alleviate the particulate $NO_3^-$ peaks during severe winter haze episodes was seldom reported. Only Guo et al. (2018b) used a thermodynamic model to estimate the sensitivity of particulate $NO_3^-$ to TA (sum of ammonia and ammonium) during one winter haze episode in Beijing. In their study, the atmospheric chemistry simulations based on $NH_3$ emission controls scenario were lacking to demonstrate the regional effects.

To alleviate severe particulate $NO_3^-$ pollution in northern China is urgent, the study on the effectiveness by $NH_3$ emission controls is necessary. In this study, we firstly compile a comprehensive $NH_3$ emission inventory for northern China in winter of 2015, and estimate the $NH_3$ emission reductions by improving manure management. Then, the ISORROPIA-II and WRF-Chem models are used to investigate the effectiveness of $NH_3$ emission reductions on alleviating particulate $NO_3^-$ during a severe haze episode. The molar ratio based on observations is used to explore the efficiency of particulate $NO_3^-$ reductions during the severe haze conditions in wintertime."

**Methods**

1. *Line 83: the authors said the measurements were conducted in December 2015 and December 2016. Why are the results of December 2016 not shown in the paper, and why the validation was only performed in December 2015 (Fig. S2)?*

Response: In section 3.3, the analysis of the molar ratio (R) have included all observations of December 2015 and 2016. Figure S2 shows the validation of the WRF-Chem simulation during the haze episode (from 6 to 10, December 2015), since WRF-Chem does not simulate other periods.

2. *Line 86: HCl (rather than HCI).*

Response: Accepted. Revised at line 97.

3. *Line 96-110: The validation of the livestock $NH_3$ emission products should be described in detail.*

Response: Accepted. As you suggested, we added more descriptions about the validation of the livestock $NH_3$ emission products in Section 2.2.

Revision: (Page 4, 120-141) "In the past few years, our inventory has been compared

with many studies to prove its reliability. For example, the spatial pattern of $NH_3$ emissions calculated in our inventory agreed well with the distribution of the $NH_3$ column concentrations in eastern Asia retrieved from the satellite measurements of Infrared Atmospheric Sounding Interferometer (IASI) (Van Damme et al., 2014). Specially, our estimation of livestock $NH_3$ emissions in China is comparable to the results of Streets et al. (2003) and Ohara et al. (2007).

Another method for estimating $NH_3$ emissions is the inverse modeling method, which provides top-down emission estimates through optimizing comparisons of model simulations with measurements. For example, Paulot et al. (2014) used the adjoint of a global chemical transport model (GEOS-Chem) and data of $NH_4^+$ wet deposition fluxes to optimize $NH_3$ emissions estimation in China. Zhang et al. (2018a) applied TES satellite observations of $NH_3$ column concentration and GEOS-Chem to provide top-down constraints on $NH_3$ emissions in China. Their estimates are 10.2 Tg $a^{-1}$ and 11.7 Tg $a^{-1}$ respectively, which are close to our results (9.8 Tg $a^{-1}$) (Paulot et al., 2014;Zhang et al., 2018a)."

**Results**

1. *Line 61: "On the one hand, the proportion of intensive livestock husbandry in China is only about 40%, far lower than that of developed countries". What's the proportion of intensive livestock husbandry in developed countries (90% or 100%)? At least, a reference should be given here.*

**Response:** Accepted. Related reference has been added.

**Revision:** (Page 5, 200-202) "… is only about 40%, far lower than that of developed countries (Harun and Ogneva-Himmelberger., 2013). As a result, the widespread free-range and grazing animal rearing …"

2. *Lines 165-170: these statements are very biased since their study timespan concerned the winter time (December), while the N application commonly occurred in spring or summer. The authors should focus on the timespan of their study, and avoid overstatements of their findings.*

**Response:** We agree with this comment. The studies quoted here are to show the backwardness of current livestock management in China. For winter, the emission reduction measures mainly focus on in-house handling and storage, since land application mainly occurs in spring and summer. To avoid ambiguity, we deleted this sentence.

**Revision:** (Page 6, Line 207-209) "… facilities for manure collection and storage (Chadwick et al., 2015).

*3. Lines 171-197: Again these statements are overstated. Actually, the authors just make a very subjective reduction in livestock NH₃ emissions, and then drive the WRF model using the reduced livestock NH₃ emission.*

**Response:** We cited more articles about exploring livestock $NH_3$ emission controls in in-house handling and storage during winter. These studies show that even under low temperature conditions in winter, the $NH_3$ emission reduction measures in in-house handling and storage are still very effective. Therefore, the proportions of $NH_3$ emission reductions used in our $NH_3$ emission inventory are reasonable. In addition, we removed the proportion of $NH_3$ emission reductions in land application due to the lack of appropriate references. In fact, in our $NH_3$ emission inventory, the $NH_3$ emissions from manure land application only account for 5% of the $NH_3$ emissions from livestock in winter. Therefore, the removal of this part of emission reductions has little effect on the overall emission reduction ratio (Total $NH_3$ emission reductions can still reach 40%). The changes are as follows:

**Revision:** (Page 6, Line 213-215) "phases: in-house handling, storage and land application (Chadwick et al., 2011). For winter, the emission reduction measures mainly focus on in-house handling and storage, since land application mainly occurs in spring and summer."

(Page 6, Line 220-223) "emissions by about 50%-70% (Balsari. et al., 2006;Petersen et al., 2013;Hou et al., 2015;Wang et al., 2017). "

(Page 6, Line 231-233) "emission reductions mentioned above were multiplied by $NH_3$ emission factors in two phases of manure management: 50% reduction at in-house handling and 60% (the average value of 50% and 70%) reduction at storage. With these measures…"

*4. Lines 199-200: In the ISORROPIA-II simulation, 40% reduction of TA was used to reflect the effects of reducing NH₃ emissions by 40%. This process is also very subjective and has no explanation at all why the authors adopted this value. At least the author should give reference to support this process. In fact, there are numerous subjective descriptions in the main text, and it's hard to specify all of them and prove them validate.*

**Response:** Accepted. As you suggested, we cited some relevant studies that used this method. We also used WRF-Chem to examine this method. Results showed that there was little difference between $NH_3$ emission reductions and TA

reductions (40% versus 40.7%). ISORROPIA-II is a box model, which calculates the thermodynamic equilibrium between aerosol phase and gas phase. It will redistribute $NH_4^+$ and $NH_3$ into aerosol phase and gas phase when TA changes. In fact, chemical transport models (e.g., WRF-Chem) also have a similar thermodynamic equilibrium calculation process when $NH_3$ emissions decreases. We added following sentences to Section 3.2.

**Revision:** (Page 6, Line 242-245) "This approach has been used in many previous studies (Blanchard and Hidy, 2003; Vayenas et al., 2005). However, in the real atmosphere, the reductions of $NH_3$ emission are not always equal to the reductions of TA due to the regional transmission. Their differences are discussed in the WRF-Chem simulation."

(Page 8, Line 298-301) "Correspondingly, TA decreased by 40.7% (from 17.2 μg/m$^3$ to 10.2 μg/m$^3$), very close to the reductions of $NH_3$ emission (40%). This indicates that it is reasonable to use TA reductions to represent $NH_3$ emission reductions in the ISORROPIA-II simulation."

**Discussions**

1. *Lines 319-336: All these were already shown in results part, but were again repeated in the discussions. I suggest the authors re-organize the discussions sector in order to summarize their results completely, also for better comparison to some latest references.*

**Response:** Accepted. We re-organized the discussions sector as you suggested.

**Revision:** (Page 11, Line 381-396) "In this study, we found that during severe winter haze episodes, the particulate $NO_3^-$ formation is $NH_3$-limited, resulting in its high sensitivity to $NH_3$ emission reductions. Meanwhile, livestock $NH_3$ emission controls is a very efficient way to alleviate particulate $NO_3^-$ pollution during severe winter hazes. The estimations showed that the improvements in manure management of livestock husbandry could effectively reduce total $NH_3$ emissions by 40% (from 100 kiloton to 60 kiloton) in winter of northern China. It would lead to a reduction of particulate $NO_3^-$ by about 40% (averagely from 40.8 to 25.7 μg/m$^3$) during severe haze conditions.

$NO_x$ emission controls could be a more direct and effective way to reduce the particulate $NO_3^-$ than $NH_3$ emission reductions. However, in northern China, the target of $NO_x$ emission reductions is only about 25% in the 13th Five-Year Plan (2016-2020) (http://www.gov.cn/zhengce/content/2017-01/05/content_5156789.htm). Due to the dominance of free-range animal rearing systems and the lack of emission controls policies, livestock $NH_3$ emission reductions in China could be practicable. In order to control $PM_{2.5}$ pollution more effectively in northern China, measures to improve manure management in livestock urgently need to be implemented."

**References**

Li, T., Wang, H., Zhao, T. L., Xue, M., Wang, Y. Q., Che, H. Z., and Jiang, C.: The Impacts of Different PBL Schemes on the Simulation of $PM_{2.5}$ during Severe Haze Episodes in the Jing-Jin-Ji Region and Its Surroundings in China, Adv Meteorol, Artn 629587810.1155/2016/6295878, 2016.

Steeneveld, G.-J.: Current challenges in understanding and forecasting stable boundary layers over land and ice, Frontiers in Environmental Science, 2, 10.3389/fenvs.2014.00041, 2014.

Steeneveld, G. J., van de Wiel, B. J. H., and Holtslag, A. A. M.: Modeling the evolution of the atmospheric boundary layer coupled to the land surface for three contrasting nights in CASES-99, J Atmos Sci, 63, 920-935, Doi 10.1175/Jas3654.1, 2006.

---

## Author Comment (AC2) · 4 Jan 2019

Our point-by-point responses are provided below. The referees' comments are italicized.

**Response to Referee #2**

*This is a straightforward and concise analysis of the sensitivity of particle nitrate loadings to winter haze episodes in Northern China. It addresses an important question –how to effectively reduce particle loadings under conditions of very bad air quality. The authors argue that because a significant proportion of Northern China's $NH_3$ emissions during the winter come from livestock, and because current agricultural practices lead to high emissions which could be reduced relatively easily (by 60% through adopting practices more common in Europe and the U.S.), that reducing total $NH_3$ emissions by 40% in the winter in achievable. Based on this argument, the paper pursues two complimentary approaches to testing the sensitivity of particle nitrate to reductions of $NH_3$. In the first, they use thermodynamic modelling of a comprehensive observational dataset obtained from measurements at a single site. While the modelling is not perfect, especially in terms of its performance for gas phase species, the authors make the case that the model results are robust for the particle phase and thus reliable for predictions when particle mass loadings are high. By applying a consistent 40% reduction to total ammonia (TA) mass loading, they find a significant reduction in particle nitrate that grows in absolute and relative terms over the course of a 4-day haze event.*

**Response:** We thank the reviewer for the very helpful comments. Please see our point-by-point reply below.

*To take a more holistic approach, the authors also perform WRF-Chem simulations over a domain centred on Northern China, performing a base case run and one in which $NH_3$ emissions from livestock were decreased by 60%. The authors make the argument for this more sophisticated approach in part because the non-linear relationship between ammonia and nitrate could change lifetime of nitrate. The authors miss an opportunity to test whether this is true under their conditions. I would encourage them to calculate the change in total nitrate (TN) burden (and/or lifetime) as a result of changing the $NH_3$ emissions.*

**Response:** Accepted. As you suggested, we calculated the change of TN burden and relevant descriptions were added to Section 3.2.

**Revision:** (Page 8, Line 296-298) "In addition, TN was reduced by 34.1% (from 31.8 µg/m³ to 21.0 µg/m³), which was in line with the assumption in Sect 2.3."

*They should also calculate the change in TA burden (and/or lifetime) to determine in a reduction in concentration of 40% is the result. Because the WRF-Chem simulations do a relatively poor job in representing TA at the observation site, confidence in the*

*model predictions is undermined. In part 3.3, the authors use the metric of molar ratio (R) to explain under what conditions particle nitrate is sensitive to reductions in TA vs TN. It would be useful if they could place their model simulation results in the context of this framework. If the model is biased in TA (or TN) but occupies a relatively 'flat' part of the isopleth diagram, then its predictions could still be robust. But if biases in the model lead to changes in R near 1, then the predictions may not be as reliable.*

**Response:** Accepted. We calculated the change in TA burden and relevant descriptions were added to Section 3.2. Meanwhile, we improved our model performance and added discussions of the simulation biases and their impacts in Section 2.2 and 3.3, respectively. During the simulation case, the average simulated value of R is 1.3, which is equivalent to the observed value (1.3). Since WRF-Chem has a good estimation of the availability of ambient $NH_3$, its estimation of the efficiency of particulate $NO_3^-$ reductions is reliable.

**Revision:** (Page 8, Line 298-301) "Correspondingly, TA decreased by 40.7% (from 17.15 μg/m³ to 10.2 μg/m³), very close to the reductions of $NH_3$ emission (40%). This indicates that it is reasonable to use TA reductions to represent $NH_3$ emission reductions in the ISORROPIA-II simulation."

(Page 5, Line 185-193) "The performance of WRF-Chem is evaluated by comparing measured and simulated $NO_3^-$, $NH_4^+$, $SO_4^{2-}$ and TA. Specifically, the observed and simulated values are, respectively: (1) $NO_3^-$, 39.8 ± 14.7 μg/m³ versus 39.1 ± 15.6 μg/m³; (2) $NH_4^+$, 27.7 ± 8.6 μg/m³ versus 26.5 ± 11.7 μg/m³; (3) $SO_4^{2-}$, 42.4 ± 16.0 μg/m³ versus 39.7 ± 20.8 μg/m³ and (4) TA, 34.6 ± 8.5 μg/m³ versus 32.1 ± 11.0 μg/m³. The MB of these four species are -0.7, -1.2, -2.7 and -2.5 μg/m³, respectively. Simulated particulate $NO_3^-$, $NH_4^+$, $SO_4^{2-}$ and TA approximately agreed with the measurements (Figure S2). There are still some simulation biases that may affect the simulation of particulate $NO_3^-$ reductions efficiency. This is discussed in detail in Sect 3.3."

(Page 12, Line 365 - 369) "Based on the above analysis, the influence of WRF-Chem simulation biases on particulate $NO_3^-$ reduction efficiency simulation mainly depends on the simulation bias of R. During the simulation case, the average simulated value of R is 1.3, which is equivalent to the observed value (1.3). Since WRF-Chem has a good estimation of the availability of ambient $NH_3$, its estimation of the efficiency of particulate $NO_3^-$ reductions is reliable."

*Specific comments*

*In the abstract and throughout the text, the authors consistently focus on the reduction in particle nitrate loading that results from reductions in $NH_3$ emissions, but particle ammonium levels also change. While the absolute change in mass loading of ammonium will be less than nitrate due to its lower molecular weight, it would still be*

*worth it in a couple of instances to calculate and report the total reduction in $PM_{2.5}$ mass from nitrate AND ammonium.*

**Response:** In Section 3.2, we have shown the changes of $PM_{2.5}$ simulation values before and after $NH_3$ emission reductions. As you suggested, we calculated the changes of the sum of particulate $NO_3^-$ and $NH_4^+$ before and after $NH_3$ emission reductions in simulations of ISORROPIA-II and WRF-Chem. Relevant descriptions were added in Section 3.2.

**Revision:** (Page 9, Line 274-275) "The sum of particulate $NO_3^-$ and $NH_4^+$ decreased from 68.7 to 46.3 $\mu g/m^3$ (a 32.6% reduction)."

(Page 9, Line 286-287) "Meanwhile, the particulate $NH_4^+$ decreased from 16.3 to 11.7 $\mu g/m^3$ (a 28.1% reduction). The sum of particulate $NO_3^-$ and $NH_4^+$ decreased from 46.9 to 30.2 $\mu g/m^3$ (a 35.6% reduction)."

*Section 2.2 More information should be provided about the inventory. Over what geographic area are the emissions quoted for? 'North China' is referred to several times, but it would be useful to be more specific. Is the region under study the totality of the six provinces shown in Figure 2, or just the area within the blue box in Figure 2? Or the domain in Figure S3? Also, is the inventory used in this work archived and available for public access?*

**Response:** Accepted. In this study, the inventory includes six provinces mentioned in the introduction sector. To make this clearer, we added the relevant description in Section 2.2. In addition, our inventory can be accessed by contacting the corresponding author. Relevant instructions are added to the Acknowledgement.

**Revision:** (Page 3, Line 108-109) "A comprehensive $NH_3$ emission inventory of northern China (including the six provinces mentioned above) in December 2015 at a monthly and 1 km × 1 km resolution …"

*Figure 2 – I suggest adding a third panel that shows either the absolute difference between the two model runs or the percent decrease. It would be useful to see the spatial pattern of the change in nitrate.*

**Response:** Accepted. The panel as you suggested has been added to Figure 2 and the relevant description has been added in section 3.2.

**Revision:** (Page 8, Line 291-293) "In some areas with high particulate $NO_3^-$ concentrations, particulate $NO_3^-$ had been effectively reduced by more than 30 $\mu g/m^3$ (shown in Figure 2c)."

*Figure S3 – Is it kg of N in $NH_3$ or kg of $NH_3$ itself?*

**Response:** It is kg of $NH_3$ itself.